# Learning to Animate Images from A Few Videos to Portray Delicate Human Actions

## Abstract

Despite recent progress, video generative models still struggle to animate human actions from static images, particularly when handling uncommon actions whose training data are limited. In this paper, we investigate the task of learning to animate human actions from a small number of videos—16 or fewer—which is highly valuable in real-world applications like video and movie production. Few-shot learning of generalizable motion patterns while ensuring smooth transitions from the initial reference image is exceedingly challenging. We propose FLASH (**F**ew-shot **L**earning to **A**nimate and **S**teer **H**umans), which improves motion generalization by aligning motion features and inter-frame correspondence relations between videos that share the same motion but have different appearances. This approach minimizes overfitting to visual appearances in the limited training data and enhances the generalization of learned motion patterns. Additionally, FLASH extends the decoder with additional layers to compensate lost details in the latent space, fostering smooth transitions from the reference image. Experiments demonstrate that FLASH effectively animates images with unseen human or scene appearances into specified actions while maintaining smooth transitions from the reference image. The animated videos are accessible on the anonymous website[1].

## 1 Introduction

Despite substantial progress (Ho et al., 2022b; Singer et al., 2022; Zhou et al., 2022; Guo et al., 2023c; Wang et al., 2023b; Esser et al., 2023; Yin et al., 2023; Liew et al., 2023; Zhang et al., 2023a; He et al., 2023; Wu et al., 2023a; Wang et al., 2024b;a), video generative models still struggle to accurately portray human actions from static images. Even commercial AI video generators, such as Dream Machine[2] from Luma AI and KLING AI[3] from Kuaishou, encounter difficulty with this task. As shown in Figure 1, both models fail to animate actions such as balance beam jump or shooting a soccer ball from static images. This difficulty arises from the scarcity of training data that specifically depict the target action. As human actions are diverse and likely follow a long-tailed distribution, many highly recognizable human actions, such as those of a niche sport like balance beam, suffer from limited training data. The data scarcity prevents data-hungry video generative models from effectively learning such actions.

In this paper, we explore the task of learning to animate human actions from a small set of videos. Our aim is to transform a static reference image into a short video of a few seconds, which portrays a specific human action described by a textual prompt. This transformation is learned from a limited dataset containing up to 16 videos for each action class, thereby reducing the need for extensive video data collection. This capability holds the promise to reduce computational cost and broaden the application domains of video generative models; it is particularly valuable for applications like video and movie production, which needs to animate specific actors performing a wide range of actions, yet each action is only used once or twice. Under such use cases, techniques requiring many example videos for each action become cost-ineffective.

Existing image animation methods encounter considerable difficulties with this task. These approaches typically rely on large video datasets for training and primarily focus on preserving the

---

[1] https://cva2099.github.io/human_action_animation/
[2] https://lumalabs.ai/dream-machine
[3] https://www.klingai.com/

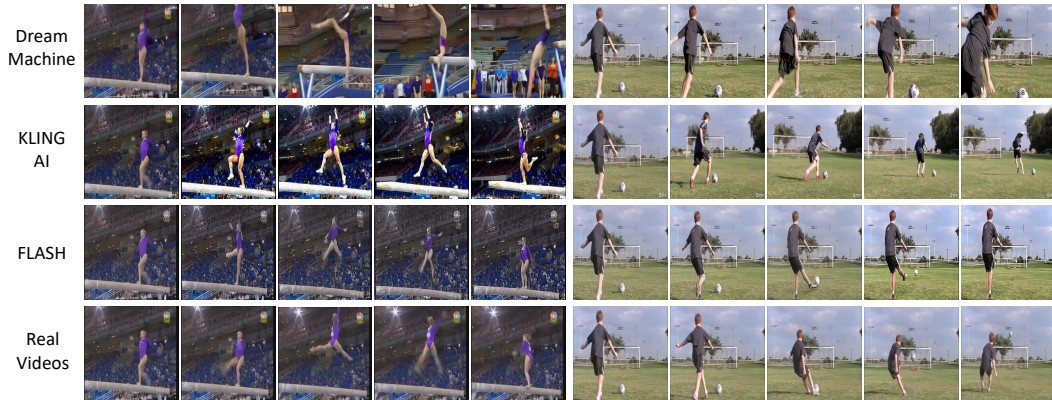

(a) An athlete is performing a balance beam jump.    (b) A person is shooting a soccer ball.

Figure 1: Comparison of animated human action videos produced by Dream Machine, KLING AI, and FLASH (our method). In the balance beam jump action, Dream Machine produces unrealistic, physics-defying movements, whereas KLING AI generates a jump but fails to portray standard jumps on the balance beam. For the soccer shooting action, both Dream Machine and KLING AI struggle to generate the correct shooting motion and the person never kicks the ball away. In contrast, FLASH successfully generates videos with higher fidelity, which resemble the real-world actions in the last row. We provide additional examples in Figure 6.

appearance of the reference images (Xing et al., 2023; Guo et al., 2023a; Jiang et al., 2023; Gong et al., 2024; Wang et al., 2023a; Guo et al., 2023b; Ma et al., 2024; Ren et al., 2024a; Gong et al., 2024; Zhang et al., 2023b) or on learning spatial-temporal conditioning controls (*e.g.*, depths or optical flows) to guide image animation (Ni et al., 2023; Kandala et al., 2024; Shi et al., 2024). However, these methods become impractical for the few-shot task. When limited to no more than 16 videos, these methods suffer from severe overfitting and fail to learn generalizable motion patterns and object transformations. Wei et al. (2024); Zhao et al. (2023) employ a two-path approach to customize motion from a few videos, but they require training for each reference image for animation, leading to limited flexibility. Although Materzynska et al. (2023); Wu et al. (2023b); Kansy et al. (2024); Li et al. (2024a) attempt to learn appearance-irrelevant motion patterns from limited data, their models lack explicit supervision for appearance-general motion, which limits performance.

The main challenge of this few-shot task is learning generalizable motion patterns. The limited number of training videos makes it difficult to learn motion patterns that generalize to diverse appearances. Furthermore, the reference image adds an extra condition, requiring the motion to align with the spatial arrangement of humans or objects in the image to maintain smooth transitions. The few-shot learning of motion conditioned on a user-provided reference image is more challenging.

To tackle this challenge, we propose FLASH (**F**ew-shot **L**earning to **A**nimate and **S**teer **H**umans), a method for few-shot human action animation. To learn generalizable motion patterns, FLASH devise the Motion Alignment Module to align the motion features and inter-frame correspondence relations between a video and its strongly augmented variant, where the motion remains the same but the appearance differs significantly. By requiring the model to predict the two videos using the two aligned motion signals, this approach encourages learning motion patterns that can generalize across different appearances, reducing overfitting to the appearance in the limited training data. Additionally, to improve transition smoothness from the reference image, FLASH employs the Detail Enhancement Decoder to propagate the details in the reference image to generated frames, which compensates for the loss of details in the latent space in the decoding process. The overall framework of FLASH is illustrated in Figure 2 (a).

Through experiments on 12 atomic human actions selected from HAA500 (Chung et al., 2021), we demonstrate that FLASH accurately animates human actions from diverse reference images while maintaining smooth transitions. It outperforms existing image animation methods across various quantitative metrics and human evaluations, showcasing the effectiveness and superiority of FLASH. Our contributions include: (1) We tackle the practical and challenging task of few-shot human action animation, an under-explored area with significant potential for video and film production. (2) We

propose FLASH, a framework designed to learn generalizable motion patterns from limited training data. (3) Experiments on 12 atomic human actions validate the effectiveness of FLASH.

## 2 RELATED WORK

**Video Generation.** Video generation using diffusion (Ho et al., 2020; Song et al., 2020b;a) have notably surpassed methods based on GANs (Goodfellow et al., 2020), VAEs (Kingma & Welling, 2013) and flow techniques (Chen et al., 2019). Diffusion models for video generation can be broadly classified into two groups. The first group generates videos purely from textual descriptions. These methods extend advanced text-to-image generative models by integrating 3D convolutions, 3D UNets, and temporal attention modules to capture temporal dynamics in videos (Ho et al., 2022b;a; Singer et al., 2022; Zhou et al., 2022; Blattmann et al., 2023; Guo et al., 2023c; Wang et al., 2023b). To mitigate concept forgetting when training on low-quality videos, some methods use both videos and images jointly for training (Ho et al., 2022b; Chen et al., 2024). Large Language Models (LLMs) contribute by generating frame descriptions (Gu et al., 2023; Huang et al., 2024; Li et al., 2024b) and scene graphs (Fei et al., 2023) to guide the video generation. Trained on large-scale video-text datasets (Bain et al., 2021; Xue et al., 2022), these methods excel at producing high-fidelity videos. However, they typically lack control over specific frame layouts, such as object positions and human poses. To improve controllability, LLMs are used to predict control signals (Lu et al., 2023; Lian et al., 2023; Lv et al., 2024), but these signals typically offer coarse control (*e.g.*, bounding boxes) rather than fine-grained control (*e.g.*, detailed human motion or object deformation).

On top of textual descriptions, the second group of techniques benefit from additional guidance sequences, such as depth maps, motion vectors, optical flows, and bounding boxes (Esser et al., 2023; Yin et al., 2023; Liew et al., 2023; Zhang et al., 2023a; He et al., 2023; Wang et al., 2024b;a), which help control motion and frame layouts. Additionally, several techniques use existing videos as guidance to generate videos with different appearances but identical motion patterns (Wu et al., 2023a; Qi et al., 2023; Yang et al., 2023; Geyer et al., 2023; Yang et al., 2024; Zhang et al., 2023c; Ling et al., 2024; Ren et al., 2024b; Park et al., 2024; Jeong et al., 2024). However, these methods cannot create novel videos that share the same motion class with the guidance video but differ in the actual motion, such as human positions and viewing angles, which limits their generative flexibility.

**Image Animation.** Image animation involves generating videos that begin with a given reference image. Common approaches achieve this by integrating the image features into videos through cross-attention layers (Wang et al., 2023a; Xing et al., 2023; Guo et al., 2023a; Jiang et al., 2023; Gong et al., 2024), employing additional image encoders (Guo et al., 2023b; Wang et al., 2024c), or incorporating the reference image into noised videos (Zeng et al., 2023; Wu et al., 2023b; Girdhar et al., 2023; Ma et al., 2024; Ren et al., 2024a; Gong et al., 2024). Another line of methods focuses on learning structural guidance (*e.g.*, motion maps) that aligns with the reference image to guide the generation of subsequent frames (Shi et al., 2024; Ni et al., 2023; Kandala et al., 2024). However, these approaches often require extensive training videos to effectively learn motion or structure guidance. Zhao et al. (2023); Wei et al. (2024) employ a temporal path to learn motion patterns from a few videos and a spatial path to learn appearance from a reference image for animation. However, they require training for each reference image, which limits their adaptability. While Materzynska et al. (2023); Wu et al. (2023b); Kansy et al. (2024); Li et al. (2024a) are similar to our work in learning specific motion patterns from a few videos, they primarily use the reference image as an appearance condition and rely on the model to automatically prioritize motion over appearance. Without explicit supervision for appearance-general motion, their generalizability is still limited. In this paper, we propose FLASH, which learns generalizable motion from only a few videos through explicit supervision, and the learned motion can be applied to reference images that differ widely in visual attributes like human positions and texture.

## 3 FLASH: FEW-SHOT LEARNING TO ANIMATE AND STEER HUMANS

To learn generalizable motion from a limited set of training videos while maintaining smooth transition from the reference image, we propose FLASH, which features two novel components as illustrated in Figure 2. The first is the Motion Alignment Module, designed to learn robust motion patterns that generalize across different appearances, which will be explained in Sec. 3.2. The

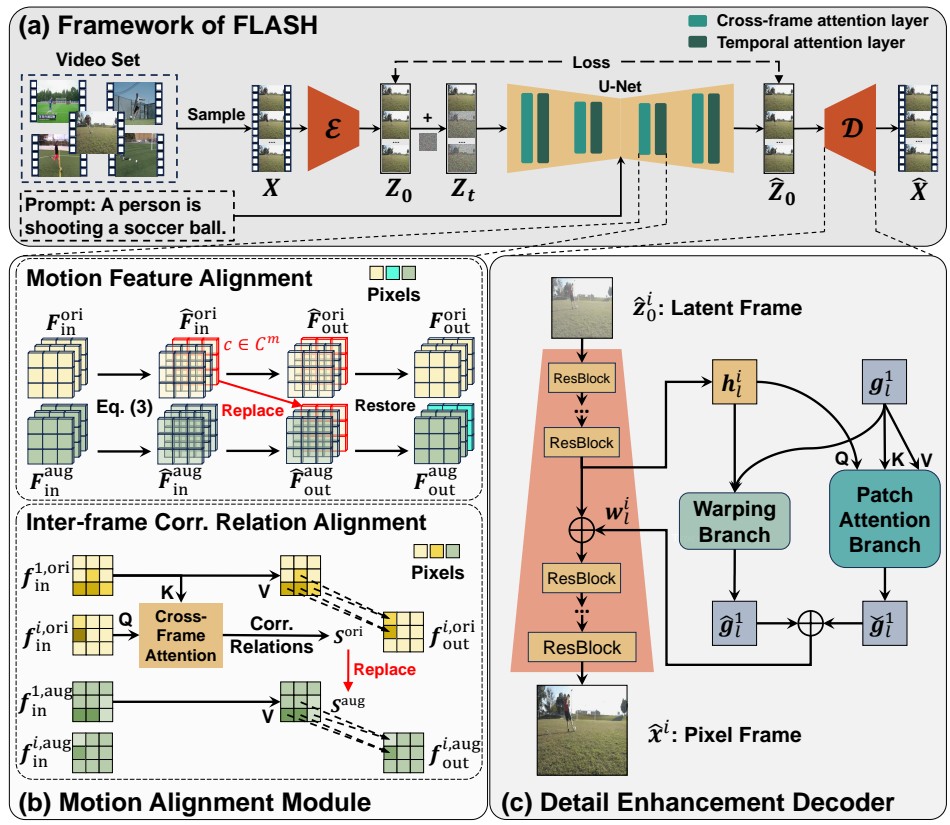

Figure 2: (a) Overview of the FLASH framework. FLASH is trained to animate human actions using a limited video set. To learn generalizable motion patterns, (b) the Motion Alignment Module aligns motion features and inter-frame correspondence relations between a training video and its strongly augmented version (see Sec. 3.2). To improve the smoothness of the transition from the reference image, (c) the Detail Enhancement Decoder propagates hierarchical details from the reference image into the generated frames (see Sec. 3.3).

second is the Detail Enhancement Decoder, which propagates details from the reference image to generated frames to enhance temporal consistency, and will be explained in Sec. 3.3.

## 3.1 PRELIMINARIES

**Image Diffusion Models.** Latent Diffusion Models (LDM) (Rombach et al., 2022), a leading image generative model, comprises four main components: an image encoder $\mathcal{E}$, an image decoder $\mathcal{D}$, a text encoder $\mathcal{T}$, and a U-Net $\epsilon_\theta$. During training, an image $\boldsymbol{x} \in \mathbb{R}^{H \times W \times 3}$ is first encoded into a latent image $\boldsymbol{z}_0 = \mathcal{E}(\boldsymbol{x}) \in \mathbb{R}^{h \times w \times c}$, where $h$, $w$ and $c$ denote the height, width and number of channels of the latent image, respectively. Next, $\boldsymbol{z}_0$ undergoes a pre-defined diffusion process (Dhariwal & Nichol, 2021; Ho et al., 2020) to add noise, resulting in $\boldsymbol{z}_t = \sqrt{\bar{\alpha}_t}\boldsymbol{z}_0 + \sqrt{1 - \bar{\alpha}_t}\boldsymbol{\epsilon}_t$, where $\boldsymbol{\epsilon}_t \sim \mathcal{N}(\mathbf{0}, I)$, $t \in [0, T]$ denotes the noising step, and $\bar{\alpha}_t$ represents the noise strength. The U-Net is then trained to predict the noise $\boldsymbol{\epsilon}_t$ from $\boldsymbol{z}_t$. During inference, a latent noise $\boldsymbol{z}_T$ is drawn from $\mathcal{N}(\mathbf{0}, I)$ and progressively denoised into $\hat{\boldsymbol{z}}_0$. Finally, the decoder reconstructs the generated image $\hat{\boldsymbol{x}} = \mathcal{D}(\hat{\boldsymbol{z}}_0)$.

**Video Diffusion Models.** The LDM framework can be naturally extended to generate videos. Given a video consisting of $N$ frames $\boldsymbol{X} = \langle \boldsymbol{x}^i \rangle_{i=1}^N$, each frame is encoded into a latent frame $\boldsymbol{z}_0^i = \mathcal{E}(\boldsymbol{x}^i) \in \mathbb{R}^{h \times w \times c}$. Collectively, all latent frames $\boldsymbol{Z}_0 = \langle \boldsymbol{z}_0^i \rangle_{i=1}^N \in \mathbb{R}^{N \times h \times w \times c}$ form a latent video used in the noising and denoising processes. The training loss is defined as:

$$\mathcal{L}_D = \mathbb{E}_{\boldsymbol{X}, \boldsymbol{\epsilon}_t \sim \mathcal{N}(\mathbf{0}, I), t, y} \left[ \|\boldsymbol{\epsilon}_t - \epsilon_\theta \left( \boldsymbol{Z}_t, t, \mathcal{T}(y) \right)\|_2^2 \right], \tag{1}$$

where $y$ is the text prompt associated with the video. To capture temporal dynamics in videos, temporal attention layers are integrated into the U-Net (Ho et al., 2022b; Esser et al., 2023; Guo et al., 2023c;b). To enhance temporal consistency between frames, the self-attention layers in the U-Net are replaced with cross-frame attention layers (Khachatryan et al., 2023; Wu et al., 2023b), in which features from the first frame (the reference frame) are used as the key and value, enabling the appearance of the first frame to be propagated to subsequent frames. In image animation tasks, the noise-free reference image is integrated into the input of the U-Net (Wu et al., 2023b; Ren et al., 2024a) to help preserve the appearance of the reference image. More details can be found in Appendix Sec. A.2.

### 3.2 MOTION ALIGNMENT MODULE

The Motion Alignment Module directs the model to learn motion that generalizes across various appearances. To achieve this, we force the model to learn consistent motion patterns from a pair of videos with identical motion but different appearances, created using strong data augmentation. We align two motion signals in the U-Net between the video pairs and requires the model to predict both videos using the shared motion signals. This approach reduces overfitting to specific appearances in limited training samples and improves generalizability of learned motion patterns. The overall process is depicted in Figure 2 (b) and explained in the following sections.

**Strongly Augmented Videos.** From the original video, $\boldsymbol{X}^{\mathrm{ori}}$, we create a strongly augmented version $\boldsymbol{X}^{\mathrm{aug}}$, which has different appearances but the same motion information. Here we choose the augmentations as Gaussian blur with random kernel sizes and random color adjustments. The overall loss is diffusion noise prediction, aimed to recover the two videos.

$$\mathcal{L}_D = \mathbb{E}_{\boldsymbol{X}^{\mathrm{ori}}, \boldsymbol{X}^{\mathrm{aug}}, \boldsymbol{\epsilon}_t^{\mathrm{ori}}, \boldsymbol{\epsilon}_t^{\mathrm{aug}}, t, y} \left[ \left\| \boldsymbol{\epsilon}_t^{\mathrm{ori}} - \epsilon_\theta \left( \boldsymbol{Z}_t^{\mathrm{ori}}, t, \mathcal{T}(y) \right) \right\|_2^2 + \left\| \boldsymbol{\epsilon}_t^{\mathrm{aug}} - \epsilon_\theta \left( \boldsymbol{Z}_t^{\mathrm{aug}}, t, \mathcal{T}(y) \right) \right\|_2^2 \right]. \quad (2)$$

For simplicity, we omit the superscripts ori and aug when the same operation is applied to both videos.

**Motion Feature Alignment.** The purpose of motion feature alignment is to force the model to learn the same motion features from the videos before and after the strong augmentation, which distorts appearance but not motion. We require the model to recover the augmented video from motion features of the original video and the appearance features of the augmented video. This encourages learning of consistent motion features from both videos. We denote the features extracted after a temporal attention layer as $\boldsymbol{F}_{\mathrm{in}} \in \mathbb{R}^{N \times h' \times w' \times c'}$. Since motion is represented by the temporal changes of the features, we remove the static components from $\boldsymbol{F}_{\mathrm{in}}$ and normalize it to obtain the dynamic features:

$$\hat{\boldsymbol{F}}_{\mathrm{in}} = \frac{\boldsymbol{F}_{\mathrm{in}} - \boldsymbol{\mu}_{\mathrm{T}}(\boldsymbol{F}_{\mathrm{in}})}{\boldsymbol{\sigma}_{\mathrm{T}}(\boldsymbol{F}_{\mathrm{in}})}, \quad (3)$$

where $\boldsymbol{\mu}_{\mathrm{T}} \in \mathbb{R}^{h' \times w' \times c'}$ and $\boldsymbol{\sigma}_{\mathrm{T}} \in \mathbb{R}^{h' \times w' \times c'}$ are the mean and standard deviation of $\boldsymbol{F}_{\mathrm{in}}$ calculated along the temporal dimension. The standard deviation serves as a normalization factor to reduce the influence of feature scales (*e.g.*, varying brightness in videos). As a result, $\hat{\boldsymbol{F}}_{\mathrm{in}}$ becomes independent of static appearance elements and is focused on the changes within the video.

However, motion information is predominantly encoded in a few channels (Xiao et al., 2024), and we need to identify the channels with rich motion information. We quantify the motion information using the standard deviations along the temporal dimension in each channel, which are then averaged across all spatial positions, and the result is denoted as $\boldsymbol{s} \in \mathbb{R}^{c'}$. Channels whose value in $\boldsymbol{s}$ exceed the $\tau$-percentile are identified as motion channels and denoted as the set $\mathcal{C}^m$. The motion features are thus represented as $\hat{\boldsymbol{F}}_{\mathrm{in}}[c], \forall c \in \mathcal{C}_m$.

We denote the motion features of the original video as $\hat{\boldsymbol{F}}_{\mathrm{in}}^{\mathrm{ori}}[c]$, and those of the augmented video as $\hat{\boldsymbol{F}}_{\mathrm{in}}^{\mathrm{aug}}[c]$. We replace $\hat{\boldsymbol{F}}_{\mathrm{in}}^{\mathrm{aug}}[c]$ with $\hat{\boldsymbol{F}}_{\mathrm{in}}^{\mathrm{ori}}[c]$ as follows:

$$\hat{\boldsymbol{F}}_{\mathrm{out}}^{\mathrm{aug}}[c] \leftarrow \hat{\boldsymbol{F}}_{in}^{\mathrm{ori}}[c], \quad \forall c \in \mathcal{C}_m. \quad (4)$$

Finally, we restore the features with video-specific mean and standard deviation, $\boldsymbol{F}_{\mathrm{out}}^{\mathrm{ori}} = \hat{\boldsymbol{F}}_{\mathrm{out}}^{\mathrm{ori}} \boldsymbol{\sigma}_{\mathrm{T}}^{\mathrm{ori}} + \boldsymbol{\mu}_{\mathrm{T}}^{\mathrm{ori}}, \boldsymbol{F}_{\mathrm{out}}^{\mathrm{aug}} = \hat{\boldsymbol{F}}_{\mathrm{out}}^{\mathrm{aug}} \boldsymbol{\sigma}_{\mathrm{T}}^{\mathrm{aug}} + \boldsymbol{\mu}_{\mathrm{T}}^{\mathrm{aug}}$, which are used in noise prediction $\epsilon_\theta(\cdot)$.

**Inter-frame Correspondence Relation Alignment.** The purpose of inter-frame correspondence relation alignment is to learn the same cross-frame motion between the original and augmented videos. From the attention weights of the original video, we identify spatial correspondence between the first frame and later frames. We then require the reconstruction of the augmented video to adopt the same spatial correspondence. This forces the diffusion model to learn the same warping strategy for both videos. Since the video pairs have the same motion but different appearance, the learned warping strategy becomes motion-sensitive and appearance-invariant.

We denote the input features of a cross-frame attention layer as $\boldsymbol{F}_{\text{in}} = \langle \boldsymbol{f}_{\text{in}}^i \rangle_{i=1}^N \in \mathbb{R}^{N \times h' \times w' \times c'}$. The output features are computed as:

$$\boldsymbol{F}_{\text{out}} = \text{CFA}(\boldsymbol{Q}, \boldsymbol{K}, \boldsymbol{V}) = \text{Softmax}\left(\frac{(\boldsymbol{Q}\boldsymbol{W}^Q)(\boldsymbol{K}\boldsymbol{W}^K)^\top}{\sqrt{c}}\right)(\boldsymbol{V}\boldsymbol{W}^V) = \boldsymbol{S}(\boldsymbol{V}\boldsymbol{W}^V), \quad (5)$$

where $\boldsymbol{Q} = \boldsymbol{F}_{\text{in}}$, $\boldsymbol{K} = \boldsymbol{f}_{\text{in}}^1$, $\boldsymbol{V} = \boldsymbol{f}_{\text{in}}^1$ are the query, key, and value, respectively, and $\boldsymbol{W}^Q$, $\boldsymbol{W}^K$, $\boldsymbol{W}^V$ are the learnable projection matrices. Unlike self-attention layers, the key and value here are from the first frame of the video provided by the user. Thus, $\boldsymbol{S}$ indicates the similarity between the query and the key from the first frame, which implicitly warps the first frame into subsequent frames (Mallya et al., 2022). Hence, $\boldsymbol{S}$ can be interpreted as correspondence relations between spatial locations of the first frame and those of the current frame, capturing cross-frame motion.

We denote the inter-frame correspondence relations of the original video and the augmented video as $\boldsymbol{S}^{\text{ori}}$ and $\boldsymbol{S}^{\text{aug}}$. We replace $\boldsymbol{S}^{\text{aug}}$ with $\boldsymbol{S}^{\text{ori}}$ in the network processing the augmented video. Effectively, this amounts to using $\boldsymbol{S}^{\text{ori}}$ to warp the features of the first frame of the augmented video to produce outputs $\boldsymbol{F}_{\text{out}}^{\text{aug}}$, which the model uses to reconstruct the augmented video. This operation enforces shared cross-frame correspondence relations (which indicate cross-frame motion) between the two videos; without learning the shared correspondence relations, the model cannot predict both videos.

### 3.3 DETAIL ENHANCEMENT DECODER

In LDM, pixel-level details can be distorted when videos are decoded from the latent space, as even minor perturbations within the latent space can lead to noticeable visual artifacts, compromising the intricate details of human actors and smooth transitions. To mitigate this issue, we devise the Detail Enhancement Decoder that extends the image decoder $\mathcal{D}$ in LDM with additional layers to propagate multi-scale details from the reference image to the generated frames. Since the motion between the reference image and generated frame can range from small to large displacements, we introduce two branches to handle both short- and long-range motion.

We define the levels of both the encoder and decoder as $l \in \{0, 1, \cdots, L\}$, with $l = 0$ representing the pixel space and $l = L$ representing the latent space. At level $l$, we extract the decoder features of the $i$-th decoding frame, denoted as $\boldsymbol{h}_l^i$, and the encoder features of the reference image, denoted as $\boldsymbol{g}_l^1$. $\boldsymbol{g}_l^1$ is then propagated to enhance the details in $\boldsymbol{h}_l^i$ through two branches, as shown in Figure 2 (c). The first branch, the warping branch, retrieves details from nearby areas in $\boldsymbol{g}_l^1$ for each spatial position in $\boldsymbol{h}_l^i$. It learns the displacements between the two features and warps $\boldsymbol{g}_l^1$ into the output $\hat{\boldsymbol{g}}_l^1$ based on these displacements. The second branch, the patch attention branch, retrieves details from the global scope of $\boldsymbol{g}_l^1$, complementing the local retrieval of the warping branch. It employs an attention layer with $\boldsymbol{h}_l^i$ as the query and $\boldsymbol{g}_l^1$ as the key and value to produce the output $\check{\boldsymbol{g}}_l^1$. The two output features are fused using learnable weights $\boldsymbol{w}_l^i$: $\tilde{\boldsymbol{h}}_l^i = \boldsymbol{h}_l^i + \boldsymbol{w}_l^i \odot (\hat{\boldsymbol{g}}_l^1 + \check{\boldsymbol{g}}_l^1)$, where $\odot$ represents element-wise multiplication. The fused features $\tilde{\boldsymbol{h}}_l^i$ is then passed to the next level. Through detail propagation at each level for each decoding frame, the details in the generated videos are enhanced.

We train the Detail Enhancement Decoder to retrieve proper details through reconstructing distorted videos to their ground-truth versions. We first extract $\boldsymbol{g}_l^1$ from the first frame of a training video. Next, we distort the video and encode it into a latent video. The decoder is then trained to reconstruct the ground-truth video using this distorted latent video. This approach encourages the decoder to retrieve relevant details from the first frame. Further details can be found in Appendix Sec. A.3.

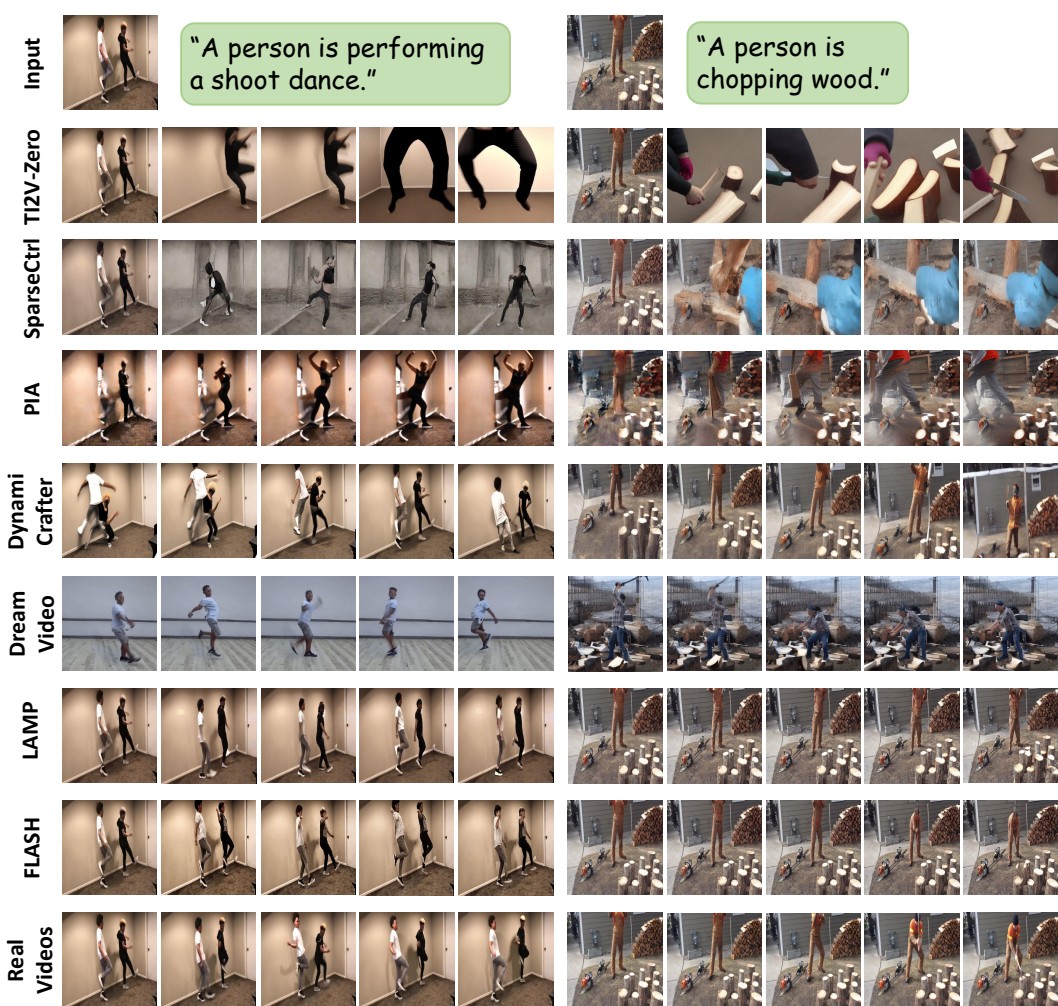

Figure 3: Qualitative comparison of different methods. Best viewed in color with zoom-in.

## 4 EXPERIMENTS

We conduct experiments on 12 actions selected from the HAA500 dataset (Chung et al., 2021). The selected actions include single-person actions (push-up, arm wave, shoot dance, running in place, and sprint run), human-object interactions (soccer shoot, drinking from a cup, balance beam jump, canoeing sprint, chopping wood, and ice bucket challenge), and human-human interactions (hugging human). More data and implementation details are described in Appendix Sec. A.4 and A.5.

### 4.1 MAIN RESULTS

**Metrics.** Following Wu et al. (2023a;b); Henschel et al. (2024), we use three CLIP-based metrics: *Text Alignment* or the similarity between generated videos and action descriptions, *Image Alignment* or the similarity between generated videos and reference images, and *Temporal Consistency* or the similarity between adjacent frames in generated videos. In these metrics, higher scores indicate better performance. Following Xing et al. (2023), we utilize Fréchet distance to compare generated videos and real ones. To mitigate content bias in the commonly used FVD (Unterthiner et al., 2018), we adopt the *CD-FVD* (Ge et al., 2024), where a lower distance indicates better performance. To assess the similarity between generated and ground-truth videos in the HAA dataset, we calculate the cosine similarity for each pair of generated and ground-truth videos. We utilize RGB and optical flow to calculate two similarity metrics: *Cosine RGB* and *Cosine Flow*. In these metrics, higher similarities reflect better performance. For all metrics, we report the average results across all test videos. More details are described in Appendix Sec. A.6.

Table 1: Quantitative comparison of different methods.

| Method | Text Alignment (↑) | Image Alignment (↑) | Temporal Consistency (↑) | CD-FVD (↓) | Cosine RGB (↑) | Cosine Flow (↑) |
|---|---|---|---|---|---|---|
| TI2V-Zero | 23.30 | 66.75 | 87.60 | 1584.30 | 0.6859 | 0.5056 |
| SparseCtrl | 21.90 | 60.77 | 88.54 | 1627.87 | 0.6704 | 0.5663 |
| PIA | 23.13 | 63.58 | 93.85 | 1547.61 | 0.6958 | 0.6055 |
| DynamiCrafter | 22.60 | **81.71** | 95.23 | 1438.01 | 0.7980 | 0.6390 |
| DreamVideo | **23.77** | 64.47 | 93.47 | 873.76 | 0.6672 | 0.6318 |
| LAMP | 22.82 | 77.93 | 93.92 | 1260.46 | 0.8284 | 0.6989 |
| FLASH | 23.02 | 79.04 | **95.64** | **786.39** | **0.8626** | **0.7786** |

**Baselines.** We compare FLASH with several image animation baselines, including the zero-shot training-free image animation model TI2V-Zero (Ni et al., 2024); large-scale trained models like SparseCtrl (Guo et al., 2023b), PIA (Zhang et al., 2023b) and DynamiCrafter (Xing et al., 2023); and motion customization models like DreamVideo (Wei et al., 2024) and LAMP (Wu et al., 2023b). More details are described in Appendix Sec. A.7.

**Qualitative Results.** We compare the qualitative performance of different methods in Figure 3. More animated videos are available on the anonymous website[1]. **TI2V-Zero** fails to create accurate or coherent actions, as it is not trained on either the target actions or the image animation task. Although **SparseCtrl**, **PIA**, and **DynamiCrafter** are trained on large-scale video datasets, they still generate unrealistic and disjointed motion that diverges considerably from the correct actions. These results reveal the limitations of large-scale pretrained video generative models in animating uncommon human actions. **DreamVideo** and **LAMP** finetune video generative models on a small set of videos containing the target actions. While DreamVideo produces realistic actions, it significantly deviates from the reference images. The results indicate that it struggles to adapt motion to different reference images flexibly, because it requires training on each reference image individually. LAMP demonstrates smooth transition from the reference image, but its rendering of the shoot dance displays discontinuities, such as disconnected or missing limbs, and it fails to generate the chopping wood action. These results demonstrate its limitations. In contrast, **FLASH** not only maintains smooth transition from the reference image but also realistically animates the intended actions that resemble real videos, demonstrating its effectiveness.

**Quantitative Results.** We compare FLASH with baselines across six metrics in Table 1. The results show that FLASH achieves the best overall performance, except in Text Alignment and Image Alignment. This suggests that FLASH generates actions with greatest temporal consistency and similarity to real action videos. In terms of Text Alignment, TI2V-Zero and DreamVideo outperform FLASH, but both exhibit significantly lower scores on Image Alignment. This implies that while they can generate correct actions, they struggle to animate reference images to portray specified actions, consistent with the qualitative results in Figure 3. In terms of Image Alignment, DynamiCrafter surpasses FLASH, but it performs considerably worse on CD-FVD, Cosine RGB, and Cosine Flow. This indicates that although DynamiCrafter maintains consistency with the reference images, it fails to generate realistic actions, as also observed in Figure 3.

**User Study.** Given the potential limitations of the CLIP, I3D, and RAFT models, we conducted a user study to further evaluate the quality of the generated videos. This study was conducted on Amazon Mechanical Turk (AMT), where workers were instructed to select the best generated video from a set of candidates. For each action, we randomly select 4 different reference images for evaluation. Control questions were included to identify random clicking, and only answers from workers who correctly answered the control questions were considered valid. More details are described in Appendix Sec. A.8. Out of 366 valid responses, FLASH was preferred in 67% of the response, significantly outperforming the next best models, DynamiCrafter (14%) and LAMP (12%). These results indicate that FLASH produces videos of the highest quality.

**Generalization to Internet and Generated Images.** To assess the generalization capability of FLASH beyond the HAA500 dataset, we tested it on images sourced from the Internet and those generated by Stable Diffusion 3 (Esser et al., 2024). As shown in Figure 4, FLASH successfully animated a variety of scenes, including a person doing a pushup in an office and running on snow. It also adapted to unrealistic scenarios, such as an astronaut running in place within a virtual space

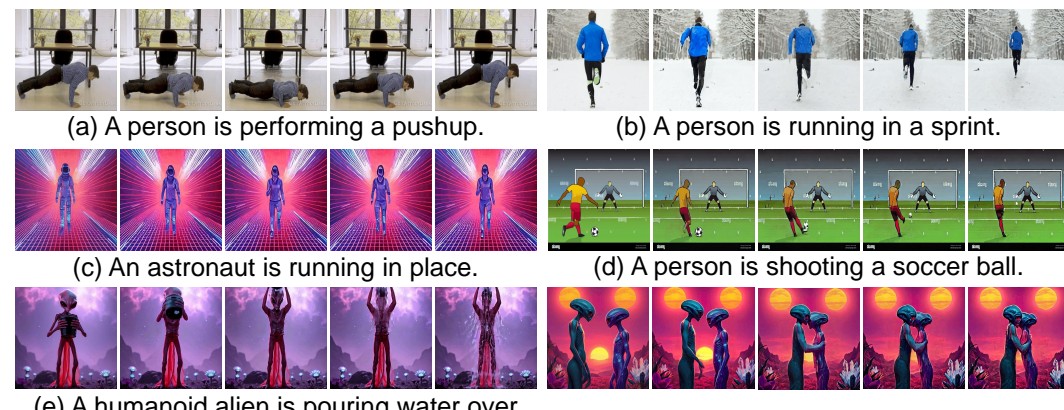

(a) A person is performing a pushup.

(b) A person is running in a sprint.

(c) An astronaut is running in place.

(d) A person is shooting a soccer ball.

(e) A humanoid alien is pouring water over his head in the ALS Ice Bucket Challenge.

(f) Two humanoid aliens are hugging.

Figure 4: Animated actions generated by FLASH using reference images sourced from the Internet and generated by image generative models.

and a cartoon character shooting a soccer ball. Additionally, FLASH can animate generated images, such as a humanoid alien pouring water over his head, two humanoid aliens hugging. More animated videos are available on the anonymous website[1]. These results highlight FLASH's strong generalization ability across a broad spectrum of reference images.

## 4.2 ABLATION STUDIES

We conducted ablation studies on four actions: sprint run, soccer shoot, canoeing sprint, and hugging human. The quantitative and qualitative results are presented in Table 2 and Figure 5, respectively. Variant #1 serves as the baseline, excluding both the Motion Alignment Module and the Detail Enhancement Decoder. Variant #2 uses only strongly augmented videos without any alignment technique. Variants #3, #4, and #5 progressively incorporate motion feature alignment, inter-frame correspondence relation alignment, and both, respectively. Lastly, Variant #6 builds upon Variant #5 by incorporating the Detail Enhancement Decoder.

Comparing the quantitative results of Variants #1 and #2, we observe that Variant #2 improves CD-FVD, Cosine RGB, and Cosine Flow, albeit with a slight decrease in CLIP scores. Qualitative results show that Variant #2 improves the fidelity of the generated actions. For example, in the soccer shooting action, the person's legs tend to disappear as the action progresses in Variant #1; however, Variant #2 preserves the leg movements. These results suggests that using augmented videos enhances the quality of generated motion.

Comparing the quantitative results of Variant #2 with Variants #3, #4, and #5, we find that Variants #3, #4, and #5 improve CD-FVD, Cosine RGB, and Cosine Flow. Both Variants #3 and #4 enhance the Cosine RGB, and Cosine Flow. When combined, Variant #5 yields further enhancements in cosine similarity and a 25-point improvements in CD-FVD, without a noticeable drop in CLIP scores. Qualitative results also indicates improved fidelity in Variants #3, #4, and #5. For instance, motion in Variant #2 appears unrealistic in both actions. In the soccer shooting action, the person's foot didn't touch the soccer ball, and the leg appears disconnected in some frames. In the canoe paddling action, the hand positions on the paddle are inconsistent across frames. However, these issues are largely mitigated in Variants #3, #4, and #5. These results demonstrate the effectiveness of the Motion Alignment Module in learning accurate motion. By providing explicit guidance for learning appearance-general motion, the module directs the model toward generalizable motion, thereby improving the quality of the generated videos.

Comparing the quantitative results of Variant #5 and Variant #6, we observe that Variant #6 noticeably improves Text Alignment and Temporal Consistency without substantially affecting CD-FVD, Cosine RGB, or Cosine Flow. Qualitatively, Variant #6 enhances some details, like the soccer ball in certain frames in the soccer shooting action, and reduces noise in generated frames. These results suggest that the Detail Enhancement Decoder could compensate for some missing details in generated frames, leading to better temporal consistency and alignment with the action descriptions.

Table 2: Quantitative ablation studies on different components of FLASH.

| Variant | Strong Augmentation | Motion Features Alignment | Inter-frame Correspondence Alignment | Detail Enhancement Decoder | Text Alignment (↑) | Image Alignment (↑) | Temporal Consistency (↑) | CD-FVD (↓) | Cosine RGB (↑) | Cosine Flow (↑) |
|---|---|---|---|---|---|---|---|---|---|---|
| #1 | ✗ | ✗ | ✗ | ✗ | 22.53 | 77.10 | 95.43 | 1023.30 | 0.8380 | 0.6806 |
| #2 | ✔ | ✗ | ✗ | ✗ | 22.48 | 76.72 | 94.91 | 932.92 | 0.8398 | 0.7061 |
| #3 | ✔ | ✔ | ✗ | ✗ | 22.64 | 76.48 | 95.06 | 920.39 | 0.8444 | 0.7140 |
| #4 | ✔ | ✗ | ✔ | ✗ | 22.70 | 76.31 | 94.84 | 938.21 | 0.8432 | 0.7172 |
| #5 | ✔ | ✔ | ✔ | ✗ | 22.52 | 76.35 | 95.01 | 906.31 | 0.8446 | 0.7224 |
| #6 | ✔ | ✔ | ✔ | ✔ | 22.77 | 76.22 | 95.31 | 908.39 | 0.8451 | 0.7233 |

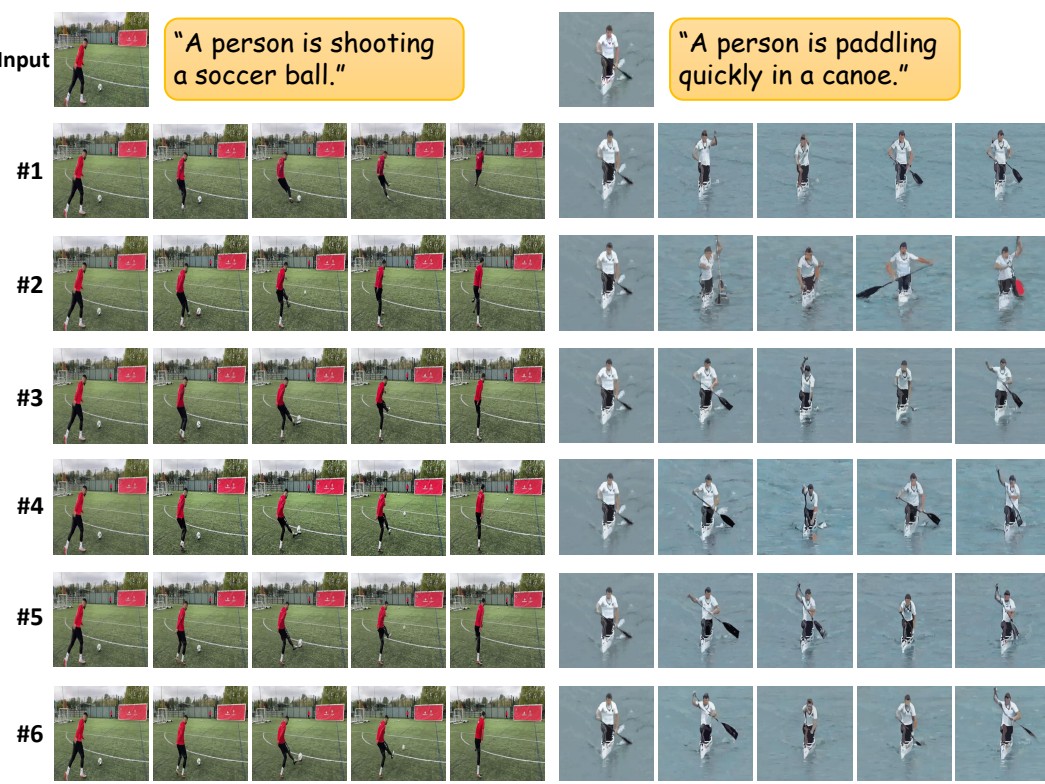

Figure 5: Qualitative ablation study on different components of FLASH.

Since the decoder operates on a frame-by-frame manner without considering inter-frame relationships when decoding, it has minimal impact on motion patterns, resulting in only slight changes on CD-FVD, Cosine RGB, and Cosine Flow.

**Applicability with Fewer Training Videos.** We examine the performance of the Motion Alignment Module in scenarios with fewer training videos (*i.e.*, 8 and 4) per action class. The results in Appendix Table 4 show that Variant #5 consistently outperforms Variant #1 and #2 in these few-shot settings, which demonstrates the ability of the Motion Alignment Module to learn generalizable motion patterns across different few-shot configurations.

**Benefits of Joint Training with Multiple Action Classes.** We evaluate whether our technique benefits from joint training with multiple action classes. We train a single model on all available videos from the four action classes. The results in Appendix Table 4 show that joint training improves nearly all metrics, particularly Image Alignment, Temporal Consistency, and Cosine RGB. The improvement indicates that joint training bolsters the performance of our technique, making it more practical for applications that require the generation of multiple delicate or customized human actions.

More ablation studies examining the effects of hyperparameters of the Motion Alignment Module and the two branches of the Detail Enhancement Module are provided in Appendix Sec. A.9.

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

# A APPENDIX

## A.1 COMPARISON OF VIDEOS GENERATED BY COMMERCIAL AI VIDEO GENERATORS

In Figure 6, we show two additional examples of human action videos generated by Dream Machine, KLING AI, and FLASH. It can be observed that Dream Machine and KLING AI fail to animate these two actions accurately. The generated videos are available on the anonymous website[1]

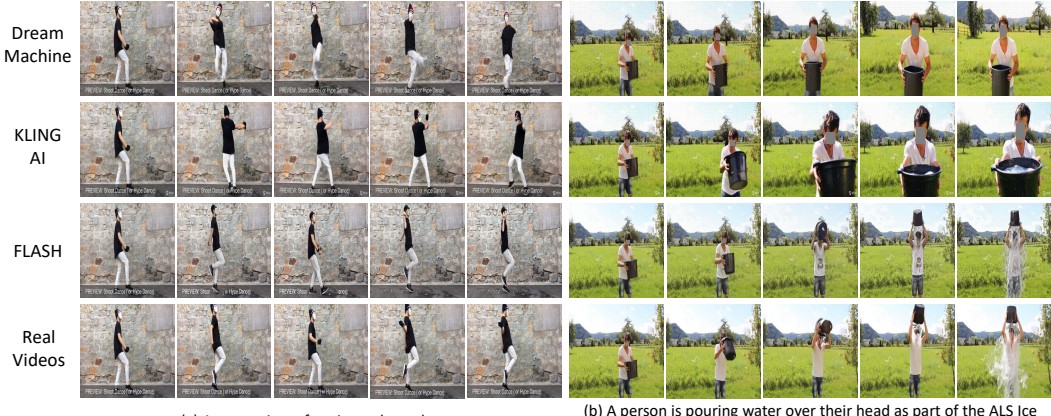

(a) A person is performing a shoot dance.

(b) A person is pouring water over their head as part of the ALS Ice Bucket Challenge.

Figure 6: Comparison of human action videos generated by Dream Machine, KLING AI, and FLASH (our method). For the shoot dance action, both Dream Machine and KLING AI produce unrealistic movements that defy physical laws. In the Ice Bucket Challenge action, neither Dream Machine nor KLING AI accurately captures the motion of pouring ice water from the bucket onto the body. In contrast, FLASH successfully generates both actions with a higher fidelity to the real movements, as shown in the last row. Human faces have been anonymized for privacy protection.

## A.2 PRELIMINARIES

**Temporal Attention Layers.** To capture temporal dynamics in videos, temporal attention layers are introduced into the U-Net (Ho et al., 2022b; Esser et al., 2023; Guo et al., 2023c;b). In a temporal attention layer, the input features $\boldsymbol{F}_{in} \in \mathbb{R}^{N \times h' \times w' \times c'}$ are first reshaped to $\tilde{\boldsymbol{F}}_{in} \in \mathbb{R}^{B \times N \times c'}$, where $B = h' \times w'$. Here, the features at different spatial locations are treated as independent samples. Temporal position encoding are then added, and a self-attention layer is applied to transform $\tilde{\boldsymbol{F}}_{in}$ into $\tilde{\boldsymbol{F}}_{out}$. Finally, $\tilde{\boldsymbol{F}}_{out}$ is reshaped back to $\boldsymbol{F}_{out} \in \mathbb{R}^{N \times h' \times w' \times c'}$ as output features. The temporal attention layer integrates information from corresponding spatial locations across frames, enabling the learning of temporal changes.

**Noise-Free Frame Conditioning.** To preserve the appearance of the reference image in the image animation task, the noise-free latent reference image is integrated into the U-Net input (Wu et al., 2023b; Ren et al., 2024a). During training, the first latent frame remains noise-free, while noise is added only to subsequent latent frames throughout the noising process. At the noising step $t$, the latent video $\boldsymbol{Z}_t = \langle \boldsymbol{z}_t^i \rangle_{i=1}^N$ is modified to $\check{\boldsymbol{Z}}_t = \langle \boldsymbol{z}_0^1, \boldsymbol{z}_t^2, \cdots, \boldsymbol{z}_t^N \rangle$, where $\boldsymbol{z}_t^1$ is replaced by $\boldsymbol{z}_0^1$. During inference, a sample $\boldsymbol{Z}_T$ is drawn from $\mathcal{N}(0, 1)$, and $\boldsymbol{z}_T^1$ is substituted with $\boldsymbol{z}_0^1 = \mathcal{E}(I)$, where $I$ is the user-provided reference image. The modified latent video $\check{\boldsymbol{Z}}_T = \langle \boldsymbol{z}_0^1, \boldsymbol{z}_T^2, \cdots, \boldsymbol{z}_T^N \rangle$ is then used for denoising. This technique effectively carries over the features from the first frame to subsequent frames, ensuring that the appearance of the reference image is preserved in the generated video.

## A.3 DETAIL ENHANCEMENT DECODER

We define the levels of both the encoder and decoder as $l \in \{0, 1, \cdots, L\}$, with $l = 0$ representing the pixel space and $l = L$ representing the latent space. At level $l$, we extract the decoder features

of the $i$-th decoding frame, denoted as $\boldsymbol{h}_l^i$, and the encoder features of the reference image, denoted as $\boldsymbol{g}_l^1$. We interpolate $\boldsymbol{g}_l^1$ to match the spatial size of $\boldsymbol{h}_l^i$ and use a fully connected layer to adjust $\boldsymbol{g}_l^1$ to the same number of channels as $\boldsymbol{h}_l^i$, resulting $\tilde{\boldsymbol{g}}_k^1$ as the input of the following two branches.

**Warping Branch.** This branch aims to retrieving details from nearby areas in $\tilde{\boldsymbol{g}}_l^1$ for each position in $\boldsymbol{h}_l^i$. It takes the channel-wise concatenation of $\boldsymbol{h}_l^i$ and $\tilde{\boldsymbol{g}}_l^1$ as input and applies four convolutional layers to estimate motion displacements from $\boldsymbol{h}_l^i$ to $\tilde{\boldsymbol{g}}_l^1$. These displacements determine the sampling positions in $\tilde{\boldsymbol{g}}_l^1$. By warping $\tilde{\boldsymbol{g}}_l^1$ based on the sampling positions, it outputs $\hat{\boldsymbol{g}}_l^1$.

**Patch Attention Branch.** This branch retrieves details from the global scope of $\tilde{\boldsymbol{g}}_l^1$, complementing the local recovery done by the warping branch. It begins by dividing both $\boldsymbol{h}_l^i$ and $\tilde{\boldsymbol{g}}_l^1$ into patches and transforming each patch into features through a fully connected layer. A cross-attention layer is then applied, using the patch features of $\boldsymbol{h}_l^i$ as the query and the patch features of $\tilde{\boldsymbol{g}}_l^1$ as the key and value, resulting in a weighted combination of $\tilde{\boldsymbol{g}}_l^1$ to produce the output $\check{\boldsymbol{g}}_l^1$.

**Feature Fusion.** To control the amount of detail added to $\boldsymbol{h}_l^i$, a two-layer convolutional network is used to learn the fusion weights. The network takes the channel-wise concatenation of $\boldsymbol{h}_l^i$ and $\tilde{\boldsymbol{g}}_l^1$ as input and outputs the fusion weights $\boldsymbol{w}_l^i$, which has the same spatial size as $\boldsymbol{h}_l^i$. The fusion is then performed as:

$$\tilde{\boldsymbol{h}}_l^i = \boldsymbol{h}_l^i + \boldsymbol{w}_l^i \odot (\hat{\boldsymbol{g}}_l^1 + \check{\boldsymbol{g}}_l^1). \tag{6}$$

The resulting feature $\tilde{\boldsymbol{h}}_l^i$ is then passed to the next level. The details in the generated frames are enhanced through the hierarchical detail propagation in each level.

**Learning to Reconstruct Distorted Videos.** We train the Detail Enhancement Decoder to retrieve proper details through reconstructing distorted videos to their ground-truth versions. During training, we first extract $\boldsymbol{g}_l^1$ using the first frame of a training video. We then intentionally distort the video using random Gaussian blur, random color adjustments on 80% of the selected areas, and random elastic transformations, and encode it into a latent video. The decoder is trained to reconstruct the ground-truth video with MSE loss. This approach encourages the decoder to retrieve relevant details from the reference image.

## A.4 DATA

We conduct experiments on 12 actions selected from the HAA500 dataset (Chung et al., 2021), which contains 500 human-centric atomic actions, each consisting of 20 short videos. The selected actions include single-person actions (push-up, arm wave, shoot dance, running in place, sprint run), human-object interactions (soccer shoot, drinking from a cup, balance beam jump, canoeing sprint, chopping wood, ice bucket challenge), and human-human interactions (hugging human).

**Training videos.** For each selected action, we use 16 videos from the training split in HAA500 for training. We manually exclude videos that contain pauses or annotated symbols in the frames. Each action label is converted into a natural sentence as the action description; for example, the action label "soccer shoot" is converted to "a person is shooting a soccer ball."

**Testing images.** For each selected action, we use the first frames from the 4 testing videos as testing images. Additionally, we search online for 2 human images depicting a person beginning the desired action as additional testing images.

## A.5 IMPLEMENTATION DETAILS

We use AnimteDiff (He et al., 2023) as the base model, initializing all parameters with its pretrained weights. The spatial resolution is set to $512 \times 512$, and the video length is set to 16 frames.

**Training of U-Net.** We use the features of the first frame and the current frame as the keys and values in the cross-frame attention layers. Noise-free frame conditioning (refer to Appendix Sec. A.2) is utilized as in Wu et al. (2023b); Ren et al. (2024a). Following Huang et al. (2023); Materzynska et al. (2023), we redefine the probability distribution for sampling denoising steps to emphasize earlier denoising stages. In the motion alignment modules, we set $\tau$ to 90 and apply motion feature alignment after each temporal attention layer in the U-Net; inter-frame correspondence relation alignment is applied to 50% of the cross-frame attention layers. For simplicity, we replace $\boldsymbol{Q}$ and

$K$ corresponding to the augmented video with those corresponding to the original video to calculate $S$, instead of replacing $S$. For Gaussian blur, we randomly sample a kernel size between 3 and 10. Color adjustments include modifications to brightness, saturation, and contrast with random factors ranging from 0.5 to 1.5, as well as hue adjustments with a random factor between -0.25 and 0.25. We only train the motion modules and the key and value projection matrices of the cross-frame attention layers. The learning rate is set to $5.0 \times 10^{-5}$, with training conducted for 20,000 steps.

**Training of Detail Enhancement Decoder.** The patch size of the Patch Attention Branch is set to 2. For video distortion, a random kernel size between 3 and 10 is used for Gaussian blur. Color adjustments involve random factors for brightness, saturation, and contrast ranging from 0.7 to 1.3, and hue adjustments ranging from -0.2 to 0.2. The displacement strength for elastic transformations is randomly sampled from 1 to 20. We only train the newly added layers, with the learning rate set to $1.0 \times 10^{-4}$ and training conducted for 10,000 steps.

**Inference.** During inference, we utilize the DDIM sampling process (Song et al., 2020a) with 25 denoising steps. Classifier-free guidance (Ho & Salimans, 2022) is applied with a guidance scale set to 7.5. Following Wu et al. (2023b), we apply AdaIN (Huang & Belongie, 2017) on latent videos for post-processing.

**Computational Resources.** Our experiments are conducted on a single GeForce RTX 3090 GPU using PyTorch, with a batch size of 1 on each GPU. We build upon the codebase of AnimateDiff (Guo et al., 2023c). Training takes approximately 36 hours per action.

## A.6 EVALUATION METRICS

In line with previous works (Wu et al., 2023a;b; Henschel et al., 2024), we use three CLIP-based metrics to assess text alignment, image alignment, and temporal consistency. (1) *Text Alignment*: We compute the similarity between each frame and the provided text prompt, averaging the scores across all frames. (2) *Image Alignment*: Similar to Text Alignment, we replace the text prompt with the provided reference image to compute the image alignment score. (3) *Temporal Consistency*: We calculate the average similarity between consecutive frame pairs to obtain the temporal consistency score. We use ViT-L-14 from OpenAI (Radford et al., 2021) for feature extraction. In these three metrics, higher scores indicate better performance.

Following Xing et al. (2023), we utilize Fréchet distance to compare generated and real videos. We use *CD-FVD* (Ge et al., 2024) to mitigate content bias in the widely used FVD (Unterthiner et al., 2018). We use VideoMAE (Tong et al., 2022), pretrained on SomethingSomethingV2 (Goyal et al., 2017), for feature extraction and distance calculation between real and generated videos. In this metric, lower distances reflect better performance.

To evaluate the similarity between generated and ground-truth videos in the HAA dataset, we calculate the cosine similarity for each pair of the generated and ground-truth videos. (1) *Cosine RGB*: We extract video features using I3D (Carreira & Zisserman, 2017), pretrained on RGB videos, for both the generated and ground truth videos, calculating cosine similarity for each pair. (2) *Cosine Flow*: We extract optical flow using RAFT (Teed & Deng, 2020) and then use I3D (Carreira & Zisserman, 2017), pretrained on optical flow data, to extract features for cosine similarity calculation. In these two metrics, higher similarities indicate better performance.

## A.7 BASELINES

We compare FLASH with several image animation baselines: (1) TI2V-Zero (Ni et al., 2024), a training-free image animation model based on a pretrained text-to-video model. (2) SparseCtrl (Guo et al., 2023b), a model trained on large-scale datasets that encodes the reference image with a sparse condition encoder and integrates the features into a video generative model. (3) PIA (Zhang et al., 2023b), a model trained on large-scale datasets that incorporates the reference image into noisy latent videos. (4) DynamiCrafter (Xing et al., 2023), a model trained on large-scale datasets that injects the reference image features into generated videos via cross-attention layers and feature concatenation. (5) DreamVideo (Wei et al., 2024), which adapts subjects and motion using a limited set of samples; we customize motion for each action using the same training videos as FLASH. (6) LAMP (Wu et al., 2023b), which learns motion patterns from a few videos; we train it with the same training videos as our method.

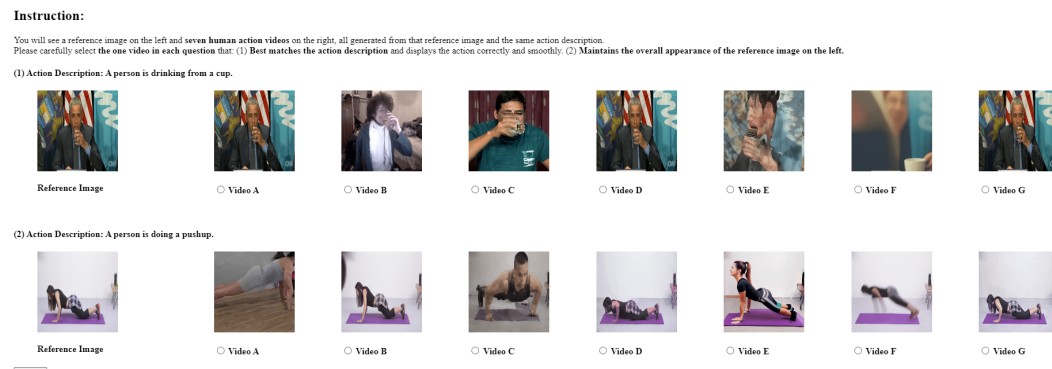

Figure 7: AMT task interface.

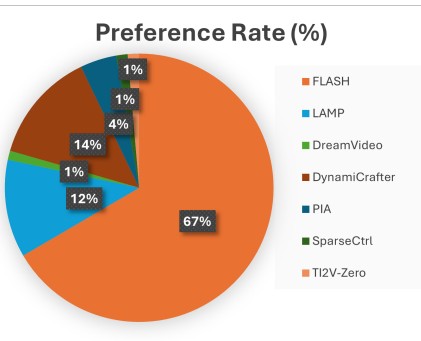

Figure 8: User preference rates (%) of different methods.

## A.8   USER STUDY

We conducted the user study on Amazon Mechanical Turk (AMT), where workers were asked to select the best-generated video from a set of candidates. For each action, 4 different reference images were randomly selected for evaluation. The AMT assessment interface is shown in Figure 7. Workers were given the following instructions: "You will see a reference image on the left and seven human action videos on the right, all generated from that reference image and the same action description. Please carefully select the one video in each question that: (1) Best matches the action description and displays the action correctly and smoothly. (2) Maintains the overall appearance of the reference image on the left." The interface also displays the reference image and the action description.

To identify random clicking, each question was paired with a control question. The control question featured a ground-truth video of a randomly selected action along with clearly incorrect videos, such as static videos or videos from the same action class that did not align with the reference image. The main and control questions were randomly shuffled to form a question pair, and each pair was evaluated by 10 different workers. Responses from workers who failed the control questions were regarded as invalid.

In total, we collected 366 valid responses. The preference rates for different methods are presented in the pie chart in Figure 8. FLASH was preferred in 67% of valid responses, substantially outperforming the next best choices, DynamiCrafter(14%) and LAMP (12%).

## A.9   ADDITIONAL ABLATION STUDIES

**Analysis of Motion Alignment Module.** In Table 3, we compare the performance of different $\tau$ values in Variant #3 and different $p$ values in Variant #4. For $\tau$, we observe that decreasing $\tau$ reduces performance in Temporal Consistency, CD-FVD, and Cosine Flow, especially in Temporal

Table 3: Ablation studies on different values of $\tau$ for motion feature alignment, different values of $p$ for motion correspondence alignment, and the impact of the warping branch and patch attention branch in the Detail Enhancement Decoder.

| Variant | $\tau$ | $p$ | Warping Branch | Patch Attention Branch | Text Alignment ($\uparrow$) | Image Alignment ($\uparrow$) | Temporal Consistency ($\uparrow$) | CD-FVD ($\downarrow$) | Cosine RGB ($\uparrow$) | Cosine Flow ($\uparrow$) |
|---|---|---|---|---|---|---|---|---|---|---|
| #3 | 90 | - | - | - | 22.64 | 76.48 | 95.06 | 920.39 | 0.8444 | 0.7140 |
| #3 | 75 | - | - | - | 22.58 | 76.63 | 95.16 | 904.25 | 0.8438 | 0.7119 |
| #3 | 50 | - | - | - | 22.57 | 77.29 | 95.14 | 934.84 | 0.8430 | 0.7031 |
| #3 | 25 | - | - | - | 22.33 | 76.52 | 94.85 | 930.53 | 0.8471 | 0.6979 |
| #4 | - | 1.0 | - | - | 22.50 | 76.43 | 94.91 | 914.12 | 0.8422 | 0.6934 |
| #4 | - | 0.5 | - | - | 22.70 | 76.31 | 94.84 | 938.21 | 0.8432 | 0.7172 |
| #5 | 90 | 0.5 | ✗ | ✗ | 22.52 | 76.35 | 95.01 | 906.31 | 0.8446 | 0.7224 |
| #6 | 90 | 0.5 | ✔ | ✗ | 22.54 | 76.21 | 95.35 | 918.61 | 0.8463 | 0.7196 |
| #6 | 90 | 0.5 | ✗ | ✔ | 22.71 | 74.97 | 95.13 | 888.05 | 0.8332 | 0.7226 |
| #6 | 90 | 0.5 | ✔ | ✔ | 22.77 | 76.22 | 95.31 | 908.39 | 0.8451 | 0.7233 |

Table 4: Analysis of training with fewer videos and joint training with multiple action classes.

| Variant | # Videos Per Class | joint Training | Text Alignment ($\uparrow$) | Image Alignment ($\uparrow$) | Temporal Consistency ($\uparrow$) | CD-FVD ($\downarrow$) | Cosine RGB ($\uparrow$) | Cosine Flow ($\uparrow$) |
|---|---|---|---|---|---|---|---|---|
| #1 | 16 | ✗ | 22.53 | 77.10 | 95.43 | 1023.30 | 0.8380 | 0.6806 |
| #2 | 16 | ✗ | 22.48 | 76.72 | 94.91 | 932.92 | 0.8398 | 0.7061 |
| #5 | 16 | ✗ | 22.52 | 76.35 | 95.01 | 906.31 | 0.8446 | 0.7224 |
| #1 | 8 | ✗ | 22.70 | 76.05 | 94.79 | 995.43 | 0.8250 | 0.6813 |
| #2 | 8 | ✗ | 22.62 | 74.37 | 94.40 | 962.82 | 0.8330 | 0.7009 |
| #5 | 8 | ✗ | 22.66 | 75.02 | 94.51 | 943.54 | 0.8340 | 0.7201 |
| #1 | 4 | ✗ | 22.22 | 72.81 | 94.24 | 1050.03 | 0.8140 | 0.6802 |
| #2 | 4 | ✗ | 22.60 | 72.00 | 93.83 | 1045.49 | 0.8188 | 0.7015 |
| #5 | 4 | ✗ | 22.46 | 72.56 | 94.22 | 1031.87 | 0.8222 | 0.7183 |
| #5 | 16 | ✗ | 22.52 | 76.35 | 95.01 | 906.31 | 0.8446 | 0.7224 |
| #5 | 16 | ✔ | 22.61 | 77.47 | 95.39 | 897.05 | 0.8501 | 0.7232 |

Consistency (94.85 for $\tau = 25$) and Cosine Flow (0.6979 for $\tau = 25$). This suggests that including more channels in motion features degrades video quality, likely because motion information is encoded in a limited number of channels (Xiao et al., 2024). Thus, we set $\tau = 90$ for the remaining experiments. Regarding $p$, substituting inter-frame correspondence relations in all cross-frame attention layers ($p = 1.0$) lowers Cosine RGB and Cosine Flow (*e.g.*, Cosine Flow drops to 0.6934 for $p = 1.0$). This might be due to the excessive regularization from substituting inter-frame correspondence relations in every layer, which makes learning difficult. Therefore, we substitute inter-frame correspondence relations in only a portion of the cross-frame attention layers.

**Analysis of Detail Enhancement Decoder.** In Table 3, we compare the effects of the Warping Branch and the Patch Attention Branch in Variant #6. Using only the Warping Branch significantly improves Temporal Consistency (from 95.01 to 95.35). In contrast, the Patch Attention Branch offers a modest gain in Text Alignment (from 22.52 to 22.71) but leads to a considerable drop in Image Alignment (from 76.35 to 74.97). Combining both branches enhances both Text Alignment and Temporal Consistency, with only a slight decrease in Image Alignment. These findings indicate that the two branches have complementary effects.

**Applicability with Fewer Training Videos.** To further assess the few-shot learning capability of the Motion Alignment Module, we conduct experiments using 8 and 4 videos randomly sampled from each action class. The results are shown in Table 4. We observe that Variant #5 consistently outperforms Variants #1 and #2 across different numbers of training videos per action class. The results validate that the Motion Alignment Module enhances the quality of animated videos in different few-shot configurations.

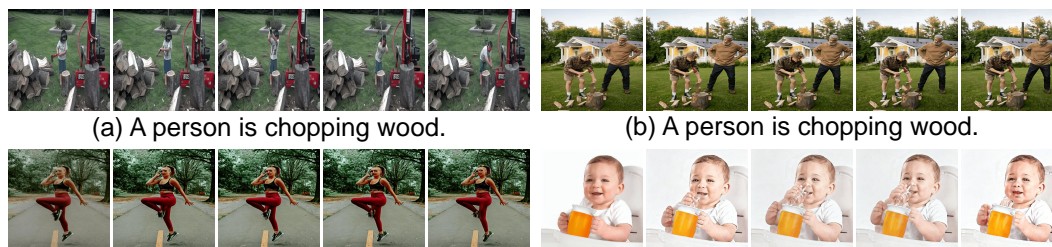

(a) A person is chopping wood.
(b) A person is chopping wood.
(c) A person is running in place.
(d) A person is drinking from a cup.

Figure 9: Failure cases.

**Joint Training with Multiple Action Classes.** We examine whether the model benefits from joint training across multiple action classes. We use the training videos from the four action classes (sprint run, soccer shoot, canoeing sprint, and hugging human) to train a single model. The results in Table 4 show improvements across nearly all metrics. The improvements in Image Alignment, Temporal Consistency, and Cosine RGB are considerable. The results suggest that joint training with multiple action classes enhances the quality of the generated videos. This makes our technique more practical for applications that need to animate images to portray multiple delicate or customized human actions.

## A.10 LIMITATIONS

Although FLASH can animate diverse reference images, it encounters challenges in accurately generating interactions involving human objects, particularly when multiple objects are present. For example, in Figure 9 (a), while a chopping action is depicted, the object being chopped is not wood. Furthermore, if the initial action status in the reference images does not align with those in the training videos, the model may struggle with animation. In Figure 9 (b), the initial action status suggests a limited range of motion for chopping wood, which differs from the training videos; in Figure 9 (c), the knee elevation motion contrasts with the steadier motion of running in place observed in the training videos; and in Figure 9 (d), a baby holding a cup with both hands deviates from the adult actions in the training videos, where one hand is used to hold the cup while drinking water. These results indicate that the model still lacks a deep understanding of motions and interactions. Employing advanced multi-modal large language models may be a promising direction to enhance the generative model's capability in addressing these challenges.

## A.11 ETHICS STATEMENT

We firmly oppose the misuse of generative AI for creating harmful content or spreading false information. We do not assume any responsibility for potential misuse by users. Nonetheless, we recognize that our approach, which focuses on animation human images, carries the risk of potential misuse. To address these risks, we are committed to maintaining the highest ethical standards in our research by complying with legal requirements and protecting privacy. Moreover, we suggest that implementing an additional content safety mechanism, similar to the one used in Stable Diffusion Rombach et al. (2022), could be an effective way to mitigate these concerns.

## A.12 CONCLUSION

In this paper, we present FLASH, a model that animates images to depict human actions using minimal training data. We employ the Motion Alignment Module to learn consistent motion signals between videos with identical motion but different appearances, facilitating the learning of generalizable motion patterns. Additionally, we introduce the Detail Enhancement Decoder to enrich details in generated videos. Experimental results show that FLASH effectively animates images with unseen human or scene appearances into specified actions while maintaining smooth transitions.

