# OpenReview forum: "Learning to Animate Images from A Few Videos to Portray Delicate Human Actions"
_ICLR.cc/2025/Conference — ICLR 2025 Conference Withdrawn Submission_

### Official Review · Reviewer_XqUA · 2024-10-30

**Soundness:** 2
**Presentation:** 2
**Contribution:** 2
**Rating:** 5
**Confidence:** 3

**Summary:**

This paper studies the problem of learning rarely-seen human motion information for video diffusion models from sparse video clips. The authors propose a "motion alignment module" to solve this challenging problem, where they first conduct pixel-level data augmentation and then encourage the model to reconstruct the video based on the shared motion information. Experiments are conducted with comparisons to baselines to show superiority of the proposed method.

**Strengths:**

- The problem studied in this paper is interesting, important, and challenging. Video diffusion models indeed manifest strong hallucinations in human motion, which need to be addressed to facilitate several downstream tasks.
- The solution of using some data augmentation techniques to solve the data sparsity issue is well-motivated. Based on this motivation, the authors design some reasonable network architecture modifications to learn from those augmented data.
- Both quantitative and qualitative metrics are shown to have a better comparison between the proposed method and the baselines. Although not beating all baselines quantitatively, the user study shows better results for the proposed method.
- Several ablation studies have been conducted to show the effectiveness of the proposed components.

**Weaknesses:**

- The technical writing of this paper is not clear enough to fully explain their method. For example, it's not clear how the method performs augmentation and whether the augmentation is reasonable. More questions can be found below.
- Some potential baselines are not compared. For example, MotionDirector trains a motion LoRA to adapt the video diffusion model to rarely seen motions. How will the proposed method compare to those methods?
- Experiment results are lacking for the comparison of the proposed method and baselines on Internet videos. It is not clear whether the proposed method is also superior on generalizability.
- The ablation studies do not show a video animation result, so it's hard to tell whether the proposed components indeed help the generation quality. Also no quantitative user study is performed for the ablation study.

**Questions:**

- How does the strongly augmented video mechanism work? What exactly is the "random color adjustment" here? After performing the "strong augmentation", is the video still in the data domain?
- Is the proposed model trained from scratch or initialized from some pre-trained model? If relying on a pre-trained model, what model does it exactly fine-tune?
- What is the requirement for the training data? How much similarity do the videos of the same motion have to share? Do they need to be in a similar view (i.e., front vs side vs back)?

---

### Official Review · Reviewer_RFFt · 2024-10-31

**Soundness:** 3
**Presentation:** 3
**Contribution:** 3
**Rating:** 6
**Confidence:** 3

**Summary:**

This paper proposes FLASH (Few-shot Learning to Animate and Steer Humans), which improves motion generalization
by aligning motion features and inter-frame correspondence relations between videos with different appearances. Even with limited training data, this approach minimizes the overfitting issue to visual appearances and enhances the generalization of learned motion patterns. Experiments demonstrate that FLASH effectively animates images with unseen human or scene appearances into specified actions while maintaining smooth transitions from the reference image.

**Strengths:**

This paper investigates the compelling and essential task of learning to animate human actions from a limited number of videos.

The problem is clearly explained and well-motivated, emphasizing the need to learn generalizable motion patterns without overfitting to the appearance of training videos, while ensuring smooth transitions from the initial reference image.

To address these key requirements, the authors introduce a Motion Alignment Module that aligns motion features and inter-frame correspondence relations. Additionally, to enhance transition smoothness from the reference image, FLASH employs a Detail Enhancement Decoder.

**Weaknesses:**

The Motion Alignment Module requires generating highly augmented versions of videos to learn motion patterns across varied appearances. This process depends heavily on the quality and effectiveness of augmentations, which, if inadequate, could fail to capture essential variances in motion or introduce irrelevant features.

**Questions:**

Could you please provides failure cases of the techniques and point out certain constraints of the work. I think this part is missing.

---

### Official Review · Reviewer_5bJ4 · 2024-11-03

**Soundness:** 3
**Presentation:** 2
**Contribution:** 3
**Rating:** 5
**Confidence:** 3

**Summary:**

This paper addresses the problem of generating videos with static images as input. Instead of having an extremely large dataset, they trained a model over a few videos(16 videos) with similar motions or actions. The key idea is to extract consistent motion patterns or features from those few videos; afterward, the human video is generated by enforcing the motion patterns as well as the appearance from the input image. They trained the models on HAA500 dataset containing several different categories of motion actions.

**Strengths:**

Although video generation models have achieved impressive advances, they rely on extensive training sets and computation resources. However, human video generation, especially with large body movements, is still challenging. The idea of exploiting very few video sequences to train video generation models specifically for humans is interesting.

**Weaknesses:**

First, some of the technical details and network structure are not very clear. For example, 1) in Fig. 2 of the system overview, the text prompt is injected into the Unet, but from the caption of the figure and also from the description in the method part, it is still not clear how to get the text prompt fed into the Unet, and do we need to have a text-encoder, cross attention as in stable diffusion?  2) What is the encoder and structure in Fig.2? Do we train the entire network together with the encoder and decoder as well as Unet, or do we need first to train the encoder and decoder? 3) When selecting the 16 videos for training, what are the selection criteria? Is the selection done manually or automatically?  4) From the given network structure and description in the method part, it is still unclear how to encode the first/reference frame image into the network.

Second, from my understanding, a model will be trained for each text prompt. This is rather inefficient.

Third, some images and visual results need to be included: 1) since augmentation plays an important role in the overall design, it is better to show some augmented images. 2) Without any video included in the supp, it is rather difficult to find out whether the temporal consistency issue exists.

**Questions:**

The network details and training procedures are rather unclear. Please refer to the questions listed in the Weakness.

---

### Official Review · Reviewer_gcwY · 2024-11-05

**Soundness:** 3
**Presentation:** 3
**Contribution:** 2
**Rating:** 6
**Confidence:** 2

**Summary:**

The paper tackles the problem of animating human actors in the few-shot setting. To address the challenges, the authors propose the FLASH framework, which mainly consists of a Motion Alignment Module and a Detail Enhancement Decoder. The  effectiveness of the method is tested on 12 atomic human actions selected from HAA500.

**Strengths:**

1. I think few-shot animation of human actors is an important direction and the proposed approach offers meaningful contributions to this field.

2. The paper compares a range of baselines and demonstrates notable improvements in terms of the alignment to the reference image and smoothness of the actions.

3. The paper is overall well organized. Ablation studies are shown for each component.

**Weaknesses:**

1. For the proposed approach, I wonder if it also works on general videos beyond human actors? If not, what is the specific design in the model tailored for human actors? I think it would add more value to the paper if it also works well on general motions. Since many of the baselines are designed for general motions, I think showing its generality is important and also makes the comparison more fair.

2. Experiments: a) The visual quality is still limited, in the examples shown, there’s still clear object flickering and motion jittering. b) It is only tested on HAA500, with 12 actions. I think that more tests on different datasets are required to see the effectiveness of the method. As for the metrics in Table 1, some metrics are worse than some baselines and I think more explanation would be helpful.

3. If given more videos, would the method still outperforms the baselines? Showing a figure that illustrates the improvements relative to the number of input videos would be helpful. This would make it clearer to understand the range in which the method outperforms others.

**Questions:**

Please refer to my questions above. Overall, I feel the method shows some promising results but more evaluations might be required to assess the approach.

---

### Note · Authors · 2024-11-15

I have read and agree with the venue's withdrawal policy on behalf of myself and my co-authors.